# Mitogen-Activated Protein Kinase Pathway in Amyotrophic Lateral Sclerosis

**DOI:** 10.3390/biomedicines9080969

**Published:** 2021-08-06

**Authors:** TG Sahana, Ke Zhang

**Affiliations:** Department of Neuroscience, Mayo Clinic, Jacksonville, FL 32224, USA

**Keywords:** amyotrophic lateral sclerosis, mitogen-activated protein kinase, stress response, c-Jun N-terminal kinase, p38, extracellular signal-regulated kinase, TAR-DNA binding protein

## Abstract

Amyotrophic lateral sclerosis is a fatal motor neuron degenerative disease. Multiple genetic and non-genetic risk factors are associated with disease pathogenesis, and several cellular processes, including protein homeostasis, RNA metabolism, vesicle transport, etc., are severely impaired in ALS conditions. Despite the heterogeneity of the disease manifestation and progression, ALS patients show protein aggregates in the motor cortex and spinal cord tissue, which is believed to be at least partially caused by aberrant phase separation and the formation of persistent stress granules. Consistent with this notion, many studies have implicated cellular stress, such as ER stress, DNA damage, oxidative stress, and growth factor depletion, in ALS conditions. The mitogen-activated protein kinase (MAPK) pathway is a fundamental mitogen/stress-activated signal transduction pathway that regulates cell proliferation, differentiation, survival, and death. Here we summarize the fundamental role of MAPK in physiology and ALS pathogenesis. We also discuss pharmacological inhibitors targeting this pathway tested in pre-clinical models, suggesting their role as potential drug candidates.

## 1. Introduction

Amyotrophic lateral sclerosis (ALS) is a fatal neurodegenerative disease affecting upper and lower motor neurons, which causes muscle weakness, atrophy, paralysis, and eventually death. According to the ALS Association, ~5000 people get diagnosed with ALS every year in the United States, and the life expectancy of these people is 2–5 years after diagnosis. Many environmental and genetic factors affect the onset of the disease, including geographical location, exposure to chemicals or radiation, extensive exercise, sports, brain, or spinal cord injury, etc. [1]. Particularly, athletes and veterans are more prone to this disease. ALS is also known as Lou Gehrig’s disease in the United States, named after one of the most famous baseball players, Lou Gehrig.

ALS can be associate with frontotemporal dementia (FTD), another fatal neurodegenerative disease characterized by frontotemporal lobe degeneration. ALS and FTD can be caused by the same mutations, affect the same individuals or patients from the same family, and share the same pathology. Thus, they are considered in a common disease spectrum [2,3]. In this review, we also include FTD cases that are related to ALS.

## 2. ALS Genetics

Around 90% of the total ALS cases have no known associated genetic factors and are referred to as sporadic ALS (sALS). The other ~10% cases have a genetic predisposition for the disease and are referred to as familial ALS (fALS). To date, there are more than 20 genes whose mutations have been shown to cause ALS [4]. Mutation in four genes, namely chromosome 9 open reading frame 72 (*C9ORF72*), fused in sarcoma (*FUS*), TAR-DNA binding protein (*TARDBP*), and superoxide dismutase 1 (*SOD1*), account for 70% of fALS cases. In addition to these genes, *MATR3*, *SETX*, *SQSTM1*, *OPTN*, *ATXN2*, *TBK1*, *ANG*, *VAPB*, *VCP*, *DAO*, *TAF15*, *UBQLN2*, *DCTN1*, *UNC13A*, *HNRNPA1*, *CHCHD10*, *SCFD1*, *MOBP*, *C21ORF2*, *NEK1*, *TUBALA*, and *PFN1* have been linked to ALS [5]. Among these genetic factors, a GGGGCC hexanucleotide expansion in *C9ORF72* is the most common genetic cause of ALS, accounting for 40% of fALS and 8–10% of sALS, whereas SOD1 is the first identified ALS gene with its mutations well studied. Notably, the same mutation in *C9ORF72* is also the most common genetic cause of FTD, accounting for 25% of its familial cases.

## 3. ALS Pathology

A pathological hallmark of ~98% of ALS and 40% of FTD is the mislocalization and aggregation of the translational product of *TARDBP*, TDP-43 [6]. In addition, aggregation of SOD1, FUS, hnRNP A1, etc., has been observed in some patients [7,8,9]. It is unclear how these proteins aggregate in ALS or FTD. Interestingly, many of these proteins are components of stress granules, RNA/protein condensates assembled in cells upon cellular stress [10,11,12,13]. Importantly, stress granules can trigger the aggregation of their component proteins under prolonged stress [14], suggesting the intriguing possibility that TDP-43, FUS, etc., aggregate in ALS via stress granule assembly. Consistent with this hypothesis, disease-causing mutations in these ALS/FTD proteins have been shown to disrupt the physical properties of stress granules, and genetically or pharmacologically inhibiting stress granule assembly can suppress neurodegeneration in several cellular and animal models of ALS [13,15,16,17,18]. However, other studies also provided evidence that ALS proteins can aggregate via stress-granule-independent ways [19,20].

## 4. ALS Pathophysiology

As a complex disease, ALS has pathophysiology involving a plethora of cellular defects that are interconnected, such as impairment in RNA metabolism, protein homeostasis, nucleocytoplasmic transport, and vesicle transport, as well as ER stress, mitochondrial dysfunction, aberrant phase separation, DNA damage, oxidative stress, axonopathy, hyperexcitability, etc. It is believed that in ALS, motor neurons are selectively susceptible to a combination of these assaults, thereby leading to their degeneration [21]. In addition, non-neuronal cells, including microglia, astrocytes, and oligodendrocytes, are also believed to contribute to the disease in a non-cell-autonomous manner [22]. Apart from defects in these cells, neuromuscular junction is also impaired in ALS. Early events include nerve sprouting, synaptic remodeling which results in reinnervation. Later stage events include withdrawal of motor axons from neuromuscular junctions, i.e., denervation which causes muscle weakness and atrophy. Furthermore, defects in the skeletal muscles can also induce a retrograde signaling cascade which contributes to motor neuron death [23,24]. However, it is unclear how different pathophysiological defects relate to each other and how they drive pathogenesis and promote disease progression.

The fact that most ALS cases exhibit TDP-43 aggregation suggests converged pathophysiological mechanisms. Indeed, the aforementioned role of stress granules in ALS pathogenesis suggests the importance of cellular stress response pathways in this disease. Notably, cellular stress response pathways have been widely implicated in neurodegeneration and aging [4,5,25]. In this review, we focus on an important cellular stress response pathway, the mitogen-activated protein kinase (MAPK) pathway, and its role in ALS etiology and progression.

## 5. MAPKs in Cellular Physiology and ALS

MAPKs are the serine and threonine kinases involved in the cellular signal transduction process. They integrate extracellular and intracellular stimuli into cellular responses that regulate differentiation, proliferation, cell death, survival, and transformation. They are activated by a wide range of stimuli, including mitogens (e.g., growth factors and cytokines), stress (e.g., toxins, hypoxic, ER stress, drugs, hyperthermia, oxidative stress, and irradiations), etc. The extracellular stimuli activate the surface receptors, for example, G protein-coupled receptors (GPCR), receptor tyrosine kinase (RTK), integrins, etc., which recruit adaptor proteins such as growth factor receptor-bound protein 2 (GRB2), son of sevenless (SOS), or small G-proteins such as Ras superfamily GTPases, etc. These activated receptor complexes recruit and activate downstream effectors, e.g., the MAPK members [26].

MAPK members can be categorized into MAP kinase kinase kinase kinase (MAP4K), MAP kinase kinase kinase (MAP3K), MAP kinase kinase (MAP2K), and MAPK. These proteins function sequentially: Once activated, MAP4Ks phosphorylate and activate MAP3Ks, which subsequently phosphorylate and activate MAP2Ks, and then MAPKs. Activation of MAPK members requires dual phosphorylation of threonine and tyrosine residues. Activated MAPKs further activate downstream targets, thereby causing transcriptional, translational, or post-translational changes in cells [27] (Figure 1).

Multiple MAPK members and four tiers of kinase activity help signal integration, signal amplification, closed regulation, and spatial-temporal regulation. Notably, the MAPK pathway is evolutionarily conserved across species (Table 1), suggesting its importance in cellular physiology and making it easy to study in animal models of diseases.

Three MAPK subgroups have been identified, extracellular signal-regulated kinases (ERKs), p38, and c-Jun N-terminal kinases (JNKs). ERKs are activated by mitogens and play roles in cell proliferation and differentiation, whereas JNKs and p38 are activated by internal or external stress and play roles in cell survival and death [28,29,30]. 

### 5.1. ERK

ERKs are activated mainly by growth factors, which bind to and activate cell surface receptors, such as RTKs, GPCR, etc. Activated receptors recruit scaffold and effector proteins (e.g., Ras), which further activate the MAP3K Raf. Activated Raf phosphorylates MAP2K members include, namely, MAP kinase ERK kinase 1 and 2 (MEK1/2), which activate ERKs (Figure 1). There are several ERKs, namely ERK 1, 2, 3, 4, 5, and 7. Activated ERK phosphorylates downstream targets, which can be either cytosolic or nuclear. The most common ERK targets are MAPK interacting kinase (MNK) 1 and 2, ribosomal S6 kinase 1–4 (RSK1–4), cAMP response element-binding protein (CREB), transcriptional regulator Myc-like (c-Myc), nuclear factor-kappa B (NF-ĸB), microtubule-associated protein (MAP), Tau, and mitogen and stress-activated protein kinase (MSK) 1 and 2. The ERK pathway is involved in the activation of pro-survival factors, which result in cellular proliferation, migration, and differentiation [26,30]. 

Abnormal phosphorylation or hyperactivation of ERK is associated with ALS conditions. Motor neurons from transgenic mice and patient tissues show ERK activation. ER/proteasomal/oxidative stress inducers, such as epoxomicin, prevent ERK 1/2 localization to the nucleus, which results in the formation of cytosolic aggregates. These motor neurons also showed impaired nuclear import, loss of nuclear TDP-43, fragmentation, and subcellular localization of TDP-43. Inhibition of ERK 1/2 activity was sufficient to cause TDP-43 defects in motor neurons but not glial cells. Furthermore, the cytosolic TDP-43 aggregates do not include cytotoxic granule-associated RNA binding protein 1 (TIA1), which is also a stress granule marker, but co-localizes with abnormally phosphorylated ERK 1/2 [31]. These data suggest the possible role of stress-induced ERK activation in TDP-43 proteinopathy. Further, postmortem spinal cord specimens of sporadic ALS patients (*n* = 10) having TDP-43 and overlapping FUS pathology showed increased p-p90RSK, a downstream effector protein of ERK [32]. 

In addition to motor neurons, astrocytes and glial cells also show abnormal ERK activation. Hyperactivation of ERK in glial cells induces neurotoxicity via the non-cell-autonomous process. A study showed that loss of nuclear TDP-43 in microglial causes activation of ERK 1/2, which activates activator protein-1 (AP-1) complex to induce expression of pro-inflammatory biomolecules, namely cyclooxygenase-2 (COX-2) and prostaglandins E 2 (PGE2). However, this process was not observed in astrocytes, suggesting a microglia-initiated neurotoxic mechanism [33]. However, in transgenic mice with a SOD1 mutant, the astrocytes showed p-ERK as early as eight weeks, and the level of p-ERK increased as the disease progresses. In addition to astrocytes, hippocampal and cerebellum regions showed ERK staining that correlates with cognitive and motor function impairment, as seen in ALS and FTLD patients [34]. Indeed, ERK is known to play a role in memory formation and synaptic plasticity [35,36]. Similarly, another study in the SOD1 mutant mice model showed the activation of ERK 1/2 via purinoceptor 7 (P2X7) receptor as a result of extracellular ATP depletion in astrocytes. ERK activation is dependent on NADPH oxidase 2 (NOX2) activation, which is responsible for the generation of ROS in glial cells, thereby forming a vicious cycle in which glial mediates neuronal toxicity [37]. 

### 5.2. p38

The p38 pathway is activated by several types of internal and external stress. Inflammatory cytokines such as tumor necrosis factor-α (TNF-α) and interleukin-1 (IL-1) can activate cell surface receptors, for example, Fas ligand (FasL), which recruit multiple scaffold proteins, such as tumor necrosis factor receptor type 1-associated death domain (TRADD), death-associated protein 6 (DAXX), receptor-interacting protein (RIP), tumor necrosis factor receptor-associated factor (TRAF), etc. Once activated, these membrane-localized complexes recruit and activate MAP3K members. In addition to receptor-mediated activation of MAP3Ks, several MAP3K members can also be directly activated by stress signals, such as hypoxia, oxidative stress, X-ray, UV-radiations, etc. Growth factors, such as transforming growth factor-β (TGF-β), can activate MAP3K members, namely transforming growth factor-β-activated kinase 1 (TAK1) and TGF-beta activated kinase 1 binding protein 1 (TAB1), which further activate MAP2Ks. MKK3 and 6 are the most common MAP2K members activated by growth factors, which further activate p38. Vertebrates have four p38 isoforms, namely p38α, β, γ, and δ. p38 activated by MAP2K members translocate to the nucleus to phosphorylate serine and threonine residues of nuclear substrates, which include growth arrest and DNA damage transcription factor 153 (GADD153), Paired box protein (Pax6), c-Jun, p53, signal transducer and activator of transcription 1 (Stat1), Myc, ETS Like-1 protein 1 (Elk1), C/EBP homologous protein (CHOP), myocyte enhancer factor 2 (MEF2), activating transcription factor 2 (ATF2), ETS Proto-Oncogene 2 transcription factor (ETS2), mitogen and stress-activated protein kinase (MSK) 1 and 2, CREB, ATF1, ETS domain protein (ER81), etc. These factors are involved in the transcription of several genes, bringing about cellular responses, such as apoptosis, cytokine production, autophagy, etc. In addition to these transcription factors, other proteins, e.g., MAPK activated protein kinase (MAPKAPKs 2/3/5), are also p38 substrates [26].

In ALS, the p38 pathway is activated in motor neurons, microglia, and astrocytes. Activation of Fas receptor caused programmed cell death (PCD) of the MNs in the sensorimotor cortex of SOD1 mutant mice. The Fas ligand activates apoptosis signal-regulating kinase 1 (ASK1), a MAP3K member, which activates p38 MAPK. Activation of p38 by Fas ligand-ASK1 causes transcriptional activation of neuronal nitric oxide synthase (nNOS) selectively in motor neurons in SOD1 transgenic mice. Nitric oxide (NO) activation by p38 forms the positive feedback loop. Exogenous NO activates the Fas receptor, which further activates DAXX and, in turn, activates nNOS in SOD1 mutant cells but not in controls [38,39]. Consistent with these data, activation of p38 in motor neurons (MNs) makes them more susceptible to neurodegeneration. Motor neurons and astrocytes in SOD1-ALS patient postmortem tissues and transgenic mice showed accumulation of p38 [40,41]. Interestingly, these p38 aggregates form skein-like inclusions and stain positive for ubiquitin and phosphorylated neurofilament at the periphery [42]. In addition, this aberrant phosphorylation of neurofilament causes the formation of p38 pathological aggregates [43]. This suggests that activation of p38 in SOD1-ALS might play a major role in the formation of pathological aggregates and induction of pro-inflammatory response.

The formation of pathological aggregates linked to p38 activation has been observed in multiple ALS models. In postmortem tissues of FUS-related ALS patients, p38-positive pathological aggregates were observed [41]. In addition to the disease pathology, SOD1 and FUS mutant models showed similar overlap in the pathomechanism. The motor neurons from SOD1 mutant transgenic mice showed impaired axonal transport [44], and similar results were observed in FUS mutant conditions. In a squid model, injection of FUS mutant into squid axoplasm showed p38 activation and impaired fast axonal transport [41], thus mimicking the gain-of-toxicity of the FUS mutant. 

Besides motor neurons, microglia and astrocytes also show p38 activation. In SOD1G93A mice, cell bodies and proximal neurites show activated p38 and phosphorylated neurofilaments during the presymptomatic stage, whereas astrocytes and microglia showed these pathologies at the later stages [44]. 

Inhibition of p38 rescues multiple cellular defects, including axonal transport defects. An unbiased small-molecule screen in embryonic-stem-cell-derived motor neurons identified inhibitors of p38 to rescue axonal transport defects in these neurons. In addition, inhibition of p38 MAPKα in these neurons rescued the axonal retrograde transport defects, and inhibition of p38 in these cells prevents cell death and promotes cell survival [40]. Moreover, an independent study showed that p38 inhibition prevents apoptosis and cyclosporin-induced cell death in SOD1 mutant motor neurons [45]. Together, these studies suggest that p38 can be a target to prevent the formation of pathological aggregates and improve neuronal physiology, thereby promoting motor neuron survival. 

### 5.3. JNK

JNKs are another group of MAPK members activated in response to stress stimuli. The JNK pathway is activated similarly to p38 and shows some overlap in terms of kinase function. First, JNKs are activated by various types of stress signals, such as cytokines, UV radiations, DNA damage, and growth factor deprivation, via multiple receptors, e.g., RTKs, GPCRs, TNF receptors, etc. Once activated, these receptors recruit several scaffold proteins that activate MAP3K members, including MEK kinase 1, 2, 3, and 4 (MEKK1–4), dual leucine zipper kinase (DLK), thousand-and-one amino acid 1 (TAO1), apoptosis signal-regulated kinase 1 (ASK1), tumor progression locus 2 (TPL2), etc. Activated MAP3Ks phosphorylate MAP2K members, mainly MEK 4 and 7, which further activate JNKs by dual phosphorylation of their tyrosine and threonine residues. Once activated, JNKs can act on both cytoplasmic and nuclear substrates. Their cytoplasmic substrates include glucocorticoid receptors, cytoskeletal proteins, tumor suppressor protein p53, MAP kinase activating death domain (MADD), Tau, STAT, etc., whereas their nuclear substrates include c-Jun, ATF2, heat shock factor (HSF)1, STAT3, etc., with c-Jun being the best characterized. Activated c-Jun can homodimerize or heterodimerize with c-Fos, thus forming the AP-1 transcription factor complex [29,30,46]. 

JNK is hyperactivated in several ALS conditions. In SOD1-ALS, JNK is activated by ER stress due to protein misfolding [47]. Misfolded proteins in the ER cause the unfolded protein response, which activates an ER transmembrane protein inositol-requiring enzyme 1 (IRE1). Activated IRE1 binds to Derlin 1 on the surface of ER, recruiting TNF receptor-associated protein 2 (TRAF2) and ASK1, a MAP3K protein. Activated ASK1 inhibits the ER-associated degradation (ERAD) pathway, resulting in JNK activation [48,49]. Interestingly, Lee et al. (2016) identified that homeodomain-interacting protein kinase 2 (HIPK2) is involved in ER-stress-mediated activation of IRE1 in multiple ALS models, including both the SOD1 and TDP-43 transgenic mouse models, as well as sALS and C9orf72-related ALS patient postmortem tissues [50]. Consistent with these findings, an independent study showed that MAP4K4 activates c-Jun and causes apoptosis. Furthermore, inhibition of MAP4K4 in mice and iPSC-derived motor neurons promotes survival [51]. Besides these mechanisms, an additional mechanism has been observed to activate JNKs in a FUS-ALS Drosophila model: Loss of function of Hippo suppresses FUS-mediated toxicity, whereas overexpression of Hippo increases toxicity, suggesting that Hippo activates JNKs in FUS-ALS [52]. 

Inhibition of JNK activity has shown promising neuroprotective effects. URMC-099, an inhibitor to hepatocyte progenitor kinase-like/germinal center kinase-like kinase (HGK), rescues cyclopiazonic acid-induced cell death in human ALS motor neurons and also prevented microgliosis in mouse microglial cells. HGK is a MAP4K member that is upstream of JNK activation [53]. 

Despite supporting evidence from several studies, others showed contradicting pieces of evidence for p38 and JNK activation in ALS, suggesting a complex role of the MAPK pathway in ALS. Due to the involvement of multiple MAPK members and the crosstalk within the MAPK subgroups, understanding the role of MAPKs in disease pathomechanism is ambiguous. Stress response MAPKs, i.e., p38 and JNK, show overlap in their functions and share some common substrates. For example, c-Jun is activated by ERK, p38, and JNK. Similarly, several upstream kinases belonging to MAP2Ks and MAP3Ks (for instance, ASK1) are the common activators for both p38 and JNK. The activation of particular MAPK is specific to the stimuli and depends on the type of cells exposed to the stress signal. For example, astrocyte and microglial behave differently and activate different stress response pathways, as discussed earlier. Hence, it is essential to evaluate their role in a context-specific manner. Studies by Veglianese et al. (2006), Wengenack et al. (2004) showed selective activation of p38 but not JNK in the spinal motor neuron of SOD1 mutant mice [54,55]. However, an unbiased deep RNAseq identified AP1 enriched in motor neurons. In SOD1 mutant-induced pluripotent stem cell (iPSC)-derived motor neurons, c-Jun levels were upregulated up to five-folds, compared to the control. In addition, these MNs showed increased ERK and JNK activation. On the other hand, FUS mutant iPSC-derived motor neurons showed p38 and ERK activation but not JNK [56]. Despite the differences in the activation of Jun, increased nuclear localization of Jun protein and mRNA specifically in motor neurons compared to other neurons across multiple models of ALS emphasizes the selective vulnerability of motor neurons to stress-induced neurodegeneration [56]. In another neuronal cell model, oxidative stress by paraquat-induced localization of TDP-43 to stress granules (SG). Inhibition of p38 and ERK prevented stress granule formation and TDP-43 localization, whereas inhibition of JNK did not prevent TDP-43 pathology. However, under sodium arsenite stress conditions, JNK prevented the formation of SGs and localization of C-terminal fragments to SGs in these cells [57]. 

In addition to these, few studies show opposing roles of JNK and p38 activation in ALS disease pathogenesis. A genetic screen in Drosophila identified that Wallenda (wnd) rescued TDP-43 induced toxicity. Wnd is a fly homolog of DLK, a MAP2K member, which is upstream to both p38 and JNK. In this TDP-43 fly model, JNK is cytoprotective, whereas p38 is cytotoxic. Loss of function of p38 and gain of function of JNK increased lifespan of flies [33]. These pieces of evidence suggest the balance in p38 and JNK activation is a critical factor determining the overall cellular response to a stress signal. 

## 6. Therapeutics Targeting MAPKs in ALS

Multiple genetic factors and heterogeneity in the pathophysiology of the disease complicate the translatability of the drugs into clinical testing. Hyperphosphorylation or abnormal phosphorylation of MAPK members is linked to several ALS-associated cellular pathophysiological defects, such as oxidative stress, neuroinflammation, ER stress, protein aggregation, axonal transport disruption, etc. Hence, targeting the MAPK members will help in better management of the disease conditions. Several inhibitors to MAPK members have been tested as a potential candidate for ALS treatment. Table 2 provides the overview of pharmacological inhibitors that have been tested in multiple ALS models.

In addition to the pharmacological inhibitors directly targeting the MAPK members, other inhibitors that target the confounding factors have shown beneficial effects. Alpha-lipoic acid, an antioxidant with free radical scavenging activities, is neuroprotective in a SOD1(G85R) transgenic fly model by targeting the PI3/AKT-ERK pathway [62]. Consistent with these findings, a case report showed that a combination treatment of sodium 2,3-dimercaptopropanesulfate together with α-lipoic acid for 3 years reversed motor neuron defects in an ALS patient [63]. Besides these approaches, intravenous injection of hIGF1 in SOD1(G93A) transgenic mice prolonged their lifespan by inhibiting the p38 and JNK pathways and mitigating the myelin pathology [64]. In addition, KCHO-1, a plant-derived compound targeting the MAPK pathway, reduced oxidative stress in hSOD1(G93A) transgenic mice [65].

## 7. Conclusions and Prospects

Studies in recent years have shown that ALS pathogenesis and disease progression involve many pathophysiological defects intra- and inter-cellularly connected, depicting the contribution of both neurons and glia. Indeed, the MAPK pathway seems to play a critical role in connecting these defects, as it is (i) one of the early activated stress response pathways in response to a wide range of stress signals, (ii) able to converge diverse stimuli into unified central cellular responses, e.g., cell survival and death, and (iii) activated in both motor neurons and glial cells across multiple ALS conditions, including sALS and fALS. Consistent with this idea, many studies have shown that several MAPKs are hyperactive in ALS patients and cell or animal models, and inhibiting MAPKs protects neurons from degeneration. However, other studies argue that MAPK activation can be neuroprotective in certain conditions, possibly due to different MAPK proteins in different contexts. Indeed, many of the MAPK members are essential, and their null mutations can cause lethality in the animal. Thus, further studies are required to dissect the precise roles of MAPKs in neurodegeneration.

As part of the complexity of eukaryotic cells, MAPK pathways have been shown to interconnect with each other, as well as with many other physiological pathways and processes in cells, some of which have been implicated in ALS and other age-related diseases. However, our knowledge about these interconnections may just be the tip of the iceberg. Future studies can further explore how MAPKs regulate cellular physiology under normal and stressed conditions, a critical question for better understanding of both fundamental cell biology and pathophysiology, as well as identifying novel therapeutic targets.

Many MAPK inhibitors have been tested in preclinical or clinical trials for ALS and other neurodegenerative diseases. Despite promising results in some preclinical models, a major concern is that MAPK pathways are essential for cell survival and proliferation. Thus, it is not surprising that several tested inhibitors exhibit strong side effects. However, this does not exclude MAPK members as potential therapeutic targets, as some of these members show cell-specific expression. If one could identify a MAPK member that is specifically expressed in a subpopulation of cells affected in the disease, e.g., certain motor neurons in ALS, inhibiting this MAPK member could still hold therapeutic potential.

## Figures and Tables

**Figure 1 biomedicines-09-00969-f001:**
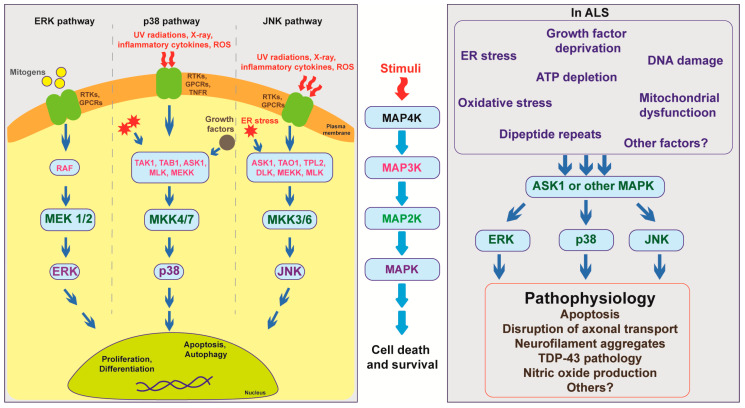
Overview of mitogen-activated protein kinase (MAPK) pathway showing the interaction of multiple MAPK members under normal physiological conditions and in ALS pathogenesis.

**Table 1 biomedicines-09-00969-t001:** List of MAPK members in human and commonly used animal models.

MAPK Members	*Homo sapiens*	*Mus musculus*	*Drosophila melanogaster*	*Caenorhabditis elegans*	*Danio rerio*
MAP4K	MAP4K1–7	Mapk4k 1/2/3/5/	Hppy, Msn		-
MAP3K	RAF (A, B, C), MOS, MEKK1–4, MLK 1/2/3/4/7, PTK, DLK, LZK, TAO, ASK, TAK1, TPL2	Raf (A, B, C), Mos, Mekk1–4, Mlk 1/2/3/4/7, Ptk, Dlk, Lzk, Tao, Ask, Tak1, Tpl2	Draf, Dmekk, Slpr, Wnd, Tao, Dask, Dtak1	Lin-45	Raf 1a/1b, c-Mos, cb135, Zak, zakl, Ask1, Tak1
MAP2K	MEK1/2, MKK3/4/5/6/7	Mek1/2, Mkk3/4/5/6/7	Lic, Dmkk4, Hep	Mek-2	Mek1/2, Mkk3/4/7
MAPK	JNK1–3, p38α/β/γ/δ, ERK1/2/3/4/5/6/7/8	Jnk1–3, p38 α/β/γ/δ, ERK1/2/3/4/5/6/7/8	Bsk, ERK-A	Mpk-1	Jnk1/2/3, Zp38a/b, Erk1/2/3/4/5/6/7

**Table 2 biomedicines-09-00969-t002:** List of MAPK inhibitors tested for ALS in various disease models.

Inhibitor	Primary Target	In Vitro or In Vivo Model	Therapeutic Effects	References
SB-239272	p38	HB9-GFP embryonic stem (ES) cells differentiated into motor neurons	Restores axonal retrograde transport deficits	[40]
MW069	p38	Squid axoplasm injected with SODG85R protein	Restores anterograde axonal transport	[58]
NQD1	ASK1	Squid axoplasm with SODG85R protein	Restores anterograde axonal transport	[58]
K811, K812	ASK1	Motor neurons differentiated from HB9-GFP cells	Rescues tunicamycin/thapsigargin induced toxicity	[59]
		SODG93A transgenic mice	Prolongs survival time of SOD mutant mice	
GNE-8505, GNE 3511	DLK	SOD1 G93A transgenic mice	Rescue axon degeneration pathology	[60]
		HB9-GFP ES cells differentiated to motor neurons	Improves MN survival and innervation of the neuromuscular junction	
SB203580 and semapimod	p38	SODG93A transgenic mouse	Increased survival of the mice and decreased axon degeneration pathology	[45]
		Primary motor neuron cultures from SOD1 mutant mice	Reduced cyclosporin induced toxicity	
		Microglial cultures from SOD mutant mice	Reduced LPS mediated activation	
Erlotinib	EGFR	SOD1G93A transgenic mouse	Delayed the onset of disease but did not extend the survival	[61]
URMC-099, prostetin/12k	MAP4K	Motor neurons from human SOD1 mutant iPSC cells	Improved motor neuron survival	[35]
		Motor neurons from mouse HB9-GFP ES cells	Improved motor neuron survival	
		N9 microglial culture	Reduced TNF induced activation	
MAP4K inhibitor 29	MAP4K	Motor neurons from human SOD1 and TDP-43 mutant iPSC	Increased motor neuron survival and reduced thapsigargin/tunicamycin induced toxicity	[51]
Trametinib	MEK	Clinical trial (phases 1 and 2)	Ongoing	Clinical trial identifier: NCT04326283

Abbreviation: ASK1-apoptosis signal-regulated kinase 1, DLK-dual leucine zipper kinase, MEK-MAPK ERK kinase, TNF-tumor necrosis factor, LPS-lipopolysaccharide, SOD-superoxide dismutase.

## Data Availability

Not applicable.

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
