# Peer review of "Mitogen-Activated Protein Kinase Pathway in Amyotrophic Lateral Sclerosis"

_biomedicines, 2021, doi:10.3390/biomedicines9080969_

Round 1

Reviewer 1 Report

In this review article “Mitogen-Activated Protein Kinase Pathway in Amyotrophic Lateral Sclerosis”, the authors summarized the fundamental role of MAPK in physiology and ALS pathogenesis, and discussed the pharmacological inhibitors targeting this pathway tested in pre-clinical models. It is a very well written and organized review article. This reviewer is impressed by the way that the authors made the complicated signal transduction pathways easy for general readers to follow. Importantly, the authors covered research results unbiasedly and provided in-depth discussions on existing contradicting results and future perspectives.       

Minor:

In Figure 1, left panel, the green text on the orange color background (plasma membrane) is harder for eyes. Suggest to use a color with better contrast.  

Author Response

Point 1: In Figure 1, left panel, the green text on the orange color background (plasma membrane) is harder for the eyes. Suggest using color with better contrast.  

Response 1: Thank you for the suggestion. We have changed the font color of the text in Figure 1 to a contrasting color making it easier for the readers to follow.

Reviewer 2 Report

This is an interesting paper that shortly summarized the physiology of MAPKs and their potential role in amyotrophic lateral sclerosis.

There is several exciting information related to MAPK subgroups and MAPK inhibitors tested for ALS. Despite the interesting data, I have comments and suggestions for Authors:

  1. The role of nerve and skeletal muscle tissues in the pathophysiology of ALS has been extensively studied and gave rise to highly inconsistent results both preclinically and clinically. For now, there is no evidence that induction of changes only in one type of tissue gives a positive change in ALS animal models. Rather a coexistence of modification/action in both types of tissue provokes positive changes in the course of ALS progression. Therefore, very interesting will also add some information focused on skeletal muscle in ALS patients/animals in the introduction, pathophysiology, and MAPK chapters.
  2. From the reader's point of view, very important is the 6th chapter: therapeutic targeting MAPKs in ALS. However, it will be great to read about data not only limited to direct inhibitors of MAPKs but also how other therapeutic factors tested in ALS act on MAPK, i.e., a-lipoic acid, other antioxidants, training, and other tested factors and their role on MAPK in the ALS model.

Author Response

Point 1: The role of nerve and skeletal muscle tissues in the pathophysiology of ALS has been extensively studied and gave rise to highly inconsistent results both preclinically and clinically. For now, there is no evidence that induction of changes only in one type of tissue gives a positive change in ALS animal models. Rather a coexistence of modification/action in both types of tissue provokes positive changes in the course of ALS progression. Therefore, very interesting will also add some information focused on skeletal muscle in ALS patients/animals in the introduction, pathophysiology, and MAPK chapters.

Response 1: We have included the possible role of non-neuronal cells including muscle cells in ALS pathophysiology in lines 76-84. Further, we have cited a recent review (reference 24) that extensively describes the role of skeletal muscle in ALS.

Point 2: From the reader's point of view, very important is the 6th chapter: therapeutic targeting MAPKs in ALS. However, it will be great to read about data not only limited to direct inhibitors of MAPKs but also how other therapeutic factors tested in ALS act on MAPK, i.e., a-lipoic acid, other antioxidants, training, and other tested factors and their role on MAPK in the ALS model.

Response 2: We agree with the reviewer. In addition to the direct inhibitors, we have added information on small molecules like antioxidants, neurotrophic factors targeting the MAPK pathway which have shown promising results in ALS models in lines 318-328.